# In Vitro Seed Germination, Seedling Development, Multiple Shoot Induction and Rooting of *Actinidia chinensis*

**DOI:** 10.3390/plants14060939

**Published:** 2025-03-17

**Authors:** Mapogo Kgetjepe Sekhukhune, Yvonne Mmatshelo Maila

**Affiliations:** 1Department of Plant Production, Soil Science and Agricultural Engineering, University of Limpopo, Polokwane 0727, South Africa; 2Limpopo Agro-Food Technology Station, University of Limpopo, Polokwane 0727, South Africa

**Keywords:** *Actinidia*, apical shoot explants, BAP, basal shoot explants, dormancy, multiple shoots, plant growth regulators, plant improvement

## Abstract

Worldwide, the yellow-fleshed kiwifruit (*Actinidia chinensis*) is an important crop that possesses great economic significance due to its nutritional, medicinal and ornamental values. The call for the expansion of the kiwifruit industry in South Africa, due to rising local and international market demand, resulted in the introduction of new plant species in sub-mountainous areas, where soil and climate conditions are more suitable for intensive kiwifruit production than in lowland areas. Consequently, a need to develop suitable commercial protocols for mass propagation of *A. chinensis* emerged. This study introduces an optimized micropropagation protocol for *A. chinensis*, facilitating seed germination, seedling development and multiple shoot induction. For seed germination, the effect of cold stratification (CS) and gibberellic acid (GA_3_) alone and in combination on in vitro germination of *A. chinensis* seeds was studied. Sterile seeds were stratified at 4 °C for 28 and 42 days. Batches of stratified and non-stratified (control) seeds were germinated on plant growth regulator-free Murashige and Skoog (MS) media and also on sterile filter paper bridges moistened with dH_2_O and GA_3_ concentrations of 500, 1000, 1500, 2000 and 2500 ppm. Seeds from the control and the CS treatments alone did not germinate on MS medium. However, on filter paper bridges, seeds cold stratified for 28 days yielded only a 20% germination percentage (GP), whereas CS for 42 days did not promote germination. A maximum GP of 64% and a mean germination time (MGT) of 27.52 days were achieved at a 2000 ppm GA_3_ concentration. Cold stratification (28 days) followed by GA_3_ treatments yielded an optimum GP of 80% and optimum MGT of 18.94 days at GA_3_ concentrations of 500 ppm. In contrast, CS (42 days) followed by GA_3_ yielded a maximum GP of 72% and MGT of 18.80 days at a GA_3_ of 500 ppm. Conclusively, CS alone had little effect on germination, whereas CS (28 and 42 days) followed by GA_3_ significantly (*p* ≤ 0.05) improved GP. Germinated seeds on moist filter paper can produce seedlings when sub-cultured on MS medium for seedling development. For multiple shoot induction, in vitro shoot culture of *A. chinensis* was carried out using apical and basal shoot explants from the above in vitro-produced seedlings. These explants were cultured on MS supplemented with 2.2 µM and 4.4 µM 6-Benzylaminopurine (BAP) for shoot multiplication. Axillary shoot proliferation was not observed on apical shoot explants after 4 weeks of culture on MS medium with 2.2 µM BAP. In contrast, the basal shoot explants produced 2–3 axillary shoots, tendrils and calluses at the base on the same medium. The highest number (3–4) of multiple shoots was attained from these basal shoot explants after subculture (10–12 weeks) in the same culture medium. In contrast, only elongation and rooting of apical shoot explants, without axillary shoot induction, occurred after the subculture. Regenerated plantlets derived from both apical and basal shoot explants were successfully acclimatised under a controlled environment at 24 ± 2 °C and 16 h photoperiod of 150–200 µmol m^−2^ s^−1^ light intensity. A similar response was observed for both types of explants of *A. chinensis* when cultured on MS with 4.4 µM BAP, although the higher concentration of BAP affected the morphological appearance of the regenerated plantlets that had shorter stems and smaller and narrower leaves compared to plantlets derived from 2.2 µM BAP.

## 1. Introduction

Kiwifruit (*Actinidia chinensis* Planch), commonly referred to as “Chinese kiwifruit”, is a woody perennial, dioecious, medicinal plant belonging to the Actinidia Lindl. genus (Actinidiaceae) comprising 75 species and about 125 taxa native to China [1]. The *A. chinensis* Planch. (yellow-fleshed) and *A. chinensis* var. *deliciosa* (green-fleshed) are categorized as two subspecies under *A. chinensis* [2]. *Actinidia chinensis* and *A. deliciosa* are by far the most important *Actinidia* species cultivated and together account for nearly all kiwifruit in international trade [3]. *Actinidia chinensis* has a special economic importance owing to its high export quality [4]. The fruit has achieved great popularity due to its commercial potential [5]. Kiwifruit is recognized as the most nutritious fruit, rich in vitamins including A, B, C, E, and K, dietary fiber, and phytochemicals such as carotenoids, flavonoids anthocyanins and lutein [6]. These essential nutrients contribute to its extensive pharmacological properties such as anti-cancerous, anti-diabetic, anti-inflammatory and antioxidant effects, as well as hypoglycemic and hypolipidemic benefits. Therefore, regular consumption of kiwifruit can enhance digestive health and support cardiovascular wellness [6]. According to FAOSTAT [7], kiwifruit is one of the most commercially available fruits on the international market with more than 4,348,011 metric tonnes of global kiwifruit production.

The South African kiwifruit (green-fleshed *A. deliciosa* ‘Hayward’) industry is gradually emerging, with approximately 200 ha under production in Limpopo, KwaZulu-Natal and Eastern Cape Provinces [8]. The commercial yield of the well-known and mostly cultivated variety of *A. delicosa* ‘Hayward’ in South Africa (SA) is limited to 10–12 t/ha. The variety of *A. delicosa* ‘Hayward’ is primarily cultivated and is targeted at local and export markets in the Southern African Development Community (SADC) countries [8]. Thus, the call to expand kiwifruit production in SA and globally calls for the development of commercial protocols for mass propagation of *A. chinensis* that can be used to expand kiwifruit production. Consequently, there is a need for a population of kiwifruit seed plants with natural or controlled crosses. Generally, the presence of dormant embryos in seeds and nonuniformity in seedling growth increase the losses of kiwifruit seed populations. Seeds are a key material both for the propagation of seedlings and breeding new kiwifruit cultivars; nevertheless, *Actinidia* has shown a high recalcitrance to seed germination, which leads to poor emergence and seedling establishment. In addition, other limitations include the impossibility of determining the sex of seedlings and genetic differentiation in generative propagation [9]. Due to the low germination performance of kiwifruit seeds, chemical treatment is required and the seeds must be kept in damp, aerated conditions, alternated between day and night, under low temperatures to increase their germination rate. According to researchers, stratification under cool and moist conditions and/or some chemical treatments can improve the germination rate of kiwifruit seeds [9,10]. Germination of kiwifruit is poor and erratic due to dormancy. The *Actinidia* seeds are known to have after-ripening (physiological) dormancy, which should be eliminated for successful germination. Physiological dormancy is the most prevalent form of dormancy in temperate seeds, which occurs when internal chemical or hormonal factors inhibit a seed from germinating, even when external conditions are favorable [11]. Previous studies suggested that low temperature and gibberellic acid can overcome kiwifruit seed dormancy [11,12,13]. Additionally, germination is a complex process that cannot be fully understood by studying one or two factors; thus, it is necessary to adopt a multidimensional strategy, combining the analysis of multiple factors to achieve optimal germination percentages. The intrinsic molecular mechanisms determining dormancy can have an embryo and/or a coat component that can interact to determine the degree of ‘whole-seed’ dormancy [11,14]. Different methods such as stratification (CS), scarification, fluctuating temperatures and gibberellic acid (GA_3_) application can be used to promote and achieve higher germination and seedling establishment. Although some studies demonstrated that GA_3_ can promote kiwifruit seed germination [9,13,15,16], other research suggests that CS is more effective in overcoming kiwifruit seed dormancy [11,14,17]. Hsieh et al. [11] achieved 80% germination in freshly harvested kiwifruit seeds using alternating temperatures (25/15 °C). Other studies highlighted the influence of both fluctuating and constant temperatures, as well as far-red and red light, on the germination of stratified ‘Hayward’ seeds. They also observed that fluctuating temperatures (20/30 °C) positively contributed to breaking kiwifruit seed dormancy. However, while red light had no effect on germination, far-red light exhibited an inhibitory effect [12].

It is also imperative to develop commercial protocols for mass propagation of *A. chinensis,* which can be used in the expansion of the SA kiwifruit production. In vitro culture is another method of mass plant production that can be used for plants that are difficult to propagate conventionally [18]. Studies on in vitro propagation of *A. chinensis* species for higher multiplication rate are limited. The first micropropagation protocol by nodal culture for *Actinidia* species was proposed by Harada [19] and it has been improved in recent years [20]. In vitro propagation of kiwifruit has also been achieved through shoot regeneration [21], organogenesis [22] and plant regeneration [23]. Explants used as initial plant material included whole leaf [24], endosperm [25], shoot tips [26], shoot meristems [20], whole buds and nodal segments [24]. However, most of these studies have primarily focused on *A. deliciosa* [22,27], *A. polygama* [20], and others with limited research on the in vitro seed culture of *A. chinensis* [28]. This is due to the fact that the treatments needed to break dormancy of kiwifruit seeds can increase the risk of contamination [20,29]. Akbaş et al. [13] studied the effect of the 6-benzylaminopurine (BAP; 0.5–4.0 mg L^−1^) on shoot proliferation of *A. deliciosa* using seeds. The optimal results were observed with 0.5 mg L^−1^ BAP, yielding an average of 4.7 ± 1.08 shoots per explant on the 4th week of culture. The in vitro-developed shoots were successfully rooted on MS medium supplemented with 1.0 mg L^−1^ α-naphthaleneacetic acid, and the resulting plantlets were successfully acclimatized to in vivo conditions. The present study was carried to investigate the effects of CS and GA_3_, both individually and in combination, on breaking dormancy and promoting germination of *A. chinensis* seeds using in vitro methods and to assess the effect of BAP on in vitro seedling development, multiple shoot induction and rooting of *A. chinensis.*

## 2. Results

### 2.1. Seed Germination

Seeds of *A. chinensis* from all CS and GA_3_ treatments and the controls exhibited poor germination when directly cultured on MS medium.

#### 2.1.1. Treatment of Seeds

Seeds cultured on moist filter paper bridges germinated, although they varied in the germination response depending on the treatment.

##### Cold Stratification

Seeds cold stratified for a mean germination time (MGT) of 28 days yielded a maximum germination percentage of 20% (GP), while those stratified for an MGT 42 days of yielded only an 8% GP.

##### Gibberellic Acid

Germination percentage and MGT were significantly (*p* ≤ 0.05) affected by GA_3_ treatments. Treatments contributed 52% total treatment variation (TTV) in GP in *A. chinensis* seeds (Table 1). Similarly, GA_3_ treatments contributed 49% TTV in MGT (Table 1).

Germination percentage

The minimum level of variable GP (8%) was observed at a GA_3_ concentration of 1500 ppm, whereas the maximum variable GP (64%) was recorded at a GA_3_ concentration of 2000 ppm. The lowest variable GPs of 4% was recorded at the control (Table 2).

Mean germination time

The minimum level of variable MGT (14.00 days) was observed at the higher GA_3_ concentration (1500 ppm), whereas the maximum MGT (30.22 days) was observed at the lower GA_3_ concentration (500 ppm).

##### Combination of 28-day Cold Stratification and GA_3_

The results for analysis of variance (ANOVA) showed that CS (28 days) followed by GA_3_ treatments significantly (*p* ≤ 0.05) enhanced germination of seeds. Treatments contributed 43% and 42% to TTV in GP and MGT in *A. chinensis* seeds, respectively (Table 3).

Germination percentage

The minimum levels of variable GP of 36% were observed at GA_3_ concentrations of 2500 ppm. The maximum variable GP of 80% was obtained at the minimum GA_3_ concentration of 500 ppm, whereas the control yielded a 20% GP (Table 4).

Mean germination time

The minimum MGT of 18.86 days was achieved at lower GA_3_ concentrations of 1000 ppm (Table 4). The maximum MGT of 26.14 days was achieved at higher GA_3_ concentrations of 2000 ppm, whereas the control had an MGT of 7.00 days. The results indicated that the MGT had strong positive correlations of 0.91 (Table 4).

##### Combination of 42-day Cold Stratification and GA_3_

The results of ANOVA revealed that GP and MGT of CS (42 days) seeds followed by GA_3_ treatments were significantly (*p* ≤ 0.05) affected. Treatments also contributed 38% and 42% TTV in GP and MGT (Table 5).

Germination percentage

At 42 days CS, low GA_3_ concentrations increased the variable GP. The minimum variable GP of 32% was achieved at GA_3_ concentrations of 2000 ppm (Table 6). The maximum variable GP of 72% was obtained at the minimum GA_3_ concentration of 500 ppm. The control yielded the lowest GP of 8% (Table 6).

Mean germination time

The minimum MGT of 15.16 days was achieved at 2000 ppm GA_3_ (Table 6). The maximum MGT of 27.30 days was achieved at 1500 ppm GA_3_ concentrations. The control was recorded at the lowest MGT of 2.80 days MGT (Table 6).

### 2.2. Seedling Development

The successfully germinated seeds on filter paper bridges, from the combined cold stratification (28- and 42-day CS) and gibberellic acid (500 ppm GA_3_) treatment (Figure 1), developed into seedlings when transferred into plant growth regulator-free full-strength Murashige and Skoog (PGR-free MS) medium (Figure 1).

Seedlings derived from seeds germinated directly on PGR-free MS medium (control) developed over a period of 9 weeks and were characterised by weaker roots, thin stems and smaller leaves exhibiting symptoms of chlorosis (Figure 2a). In contrast, seedlings derived from seeds germinated on filter paper bridges and transferred thereafter to MS culture media supplemented with 2.2 µM and 4.4 µM BAP developed over a period of 5 weeks and had well-established roots and thicker stems with bigger leaves (Figure 2b,c), when compared to the seedlings on PGR-free (control) medium (Figure 2a). Leaves from seedlings grown in MS medium supplemented with 2.2 µM BAP (Figure 2b) showed symptoms of interveinal chlorosis, whereas only mild marginal chlorosis was observed on seedlings grown on MS medium supplemented with 4.4 µM BAP (Figure 2c).

### 2.3. Explant Source

The suitability of in vitro produced seedlings of *A. chinensis* as explant sources for shoot multiplication was assessed. The in vitro-produced seedlings of *A. chinensis* were successfully used for obtaining apical (+A) and basal (−A) shoot explants for in vitro shoot culture. These explants derived from aseptic seedling cultures required no further sterilisation and showed no microbial contaminations in subsequent shoot culture.

### 2.4. Shoot Induction and Multiplication

#### 2.4.1. Effect of MS Media Supplemented with 2.2 µM BAP on Shoot Multiplication of *Actinidia chinensis*

Axillary shoot proliferation was not observed on apical (+A) shoot explants after 4 weeks of culture on MS media supplemented with 2.2 µM BAP (Figure 3a). In contrast, the basal (−A) shoot explants produced 2–3 axillary shoots, tendrils and calluses at the base of the explants (Figure 3b). Both types of explants showed expansion of the pre-existing leaves.

The subculture of apical (+A) and basal (−A) shoot explants transferred to a fresh MS medium with 2.2 µM BAP resulted in rapid leaf proliferation after only two weeks in culture. Further incubation of apical (+A) shoot explants in this culture medium for up to 10 weeks resulted in rooting and regeneration of a single plantlet per explant (Figure 3a). The basal shoot explant elongated and rooted after 10–12 weeks (Figure 3b). Multiple axillary shoots (2–3) per explant were induced on these explants grown in the presence of 2.2 µM BAP for 12 weeks, which can serve as a secondary explant source (Figure 3(b1,b2)).

#### 2.4.2. Effect of MS Media Supplemented with 4.4 µM BAP on Shoot Multiplication of *Actinidia chinensis*

In MS medium supplemented with a higher BAP concentration of 4.4 µM, no axillary shoots were induced at the apical (+A) shoot explant. Subculture of apical (+A) shoot explants transferred to fresh MS of the same composition resulted in elongation of the shoots and rooting after 10–12 weeks of culture (Figure 4a). In contrast, expansion of pre-existing leaves, proliferation of new leaves and induction of axillary shoots (2–3) were observed in basal (−A) shoot explants. Tendrils and calluses also formed at the base of this explant. Subculture of the axillary shoot derived from the basal shoot explants transferred to fresh MS medium with 4.4 µM BAP resulted in regeneration of 2–3 plantlets after 10–12 weeks of culture (Figure 4b).

### 2.5. Root Induction, Acclimatisation and Transplanting of Plantlets

The in vitro-developed shoots from 2.2 µM and 4.4 µM BAP were successfully rooted on a PGR-free MS medium over a period of 3 weeks (Figure 5). The rooted plantlets of *A. chinensis* were successfully acclimatised for 3–4 weeks in vivo inside plastic pots filled with moistened vermiculite by gradual exposure of the plants to lower humidity (Figure 5A(1,2)). Relatively higher percentages up to 70% of in vitro plantlets were obtained over a period of 11 months (Figure 5B–D(1,2)). Supplementation with nutrient solution (Multisol^®^ ‘N’) was required to eliminate symptoms of chlorosis from the 12th week onwards.

## 3. Discussion

### 3.1. In Vitro Seed Germination

Seeds of *A. chinensis* from all CS and GA_3_ treatments and the controls exhibited poor germination when directly cultured on MS medium. Typically, *Actinidia* seeds are germinated using a short cold treatment with adequate moisture, followed by stratification. However, this method is not suitable for agar medium germination due to the increased risk of contamination. There have been limited studies on the in vitro germination of *Actinidia* species such as *A. arguta*, *A. chinensis*, and *A. deliciosa* [13,30]. Wu and Datson [30] reported germination rates of up to 85% in two diploid selections of *A. chinensis* seeds treated with GA_3_ when germinated on agar medium. Additionally, the combined effect of warm water treatment (37 °C) and GA_3_ (500 ppm) resulted in a 95% GP. The poor germination observed in our study may be attributed to the composition of the culture medium and sterilization procedure. Wu and Datson [30] found that using half-strength MS medium with a low sucrose concentration (1%) promoted germination in *Actinidia* species. Esfandiari et al. [10] observed varying germination responses depending on the *Actinidia* species when seeds from CS, GA_3_, and a combination of both were germinated on water agar (7 g/L agar) in Petri dishes under two temperature conditions: a constant temperature of 24 °C or an alternating temperature cycle of 24/12 °C. The selection of an appropriate culture medium depends on the specific plant species being cultured [31], as some species are sensitive to high salt concentrations or require specific PGRs. Additionally, factors such as plant age and the type of organ being cultured play a crucial role [32]. Further research is needed to investigate the effects of MS medium strength, sucrose concentration and growth regulators on the in vitro germination of *A. chinensis* seeds to produce seedlings that can serve as an aseptic explant source for in vitro multiplication.

#### 3.1.1. Treatment of Seeds

In contrast with results from the MS medium, germination was observed in seeds cultured on moist filter paper bridges, although with various germination responses depending on the seed treatment.

##### Cold Stratification

Kiwifruit seeds are known to possess germination difficulties, which has led to extensive research studies to explore techniques that can break the endogenous dormancy embedded in seeds. The results indicated that seed germination parameters such as GP and MGT were significantly influenced by the duration of CS in breaking dormancy. According to our findings, seeds subjected to 28 days of CS achieved a maximum GP of 20%, while those stratified for 42 days reached only an 8% GP. However, these germination percentages were lower than those reported in previous studies. Maghdouri et al. [17] recorded a maximum GP of 86% in *A. chinensis* CK2 after 28 days of CS, whereas the minimum GP of 1% was observed in *A. chinensis* CK7 and CK9 after 21 days of CS. Their study also demonstrated that a longer duration of stratification generally resulted in higher germination percentages. With their *A. deliciosa* DA genotype studies, the maximum and minimum GPs belonged to *A. deliciosa* DA2 (99%) and *A. deliciosa* DA4 (34%) after four weeks of CS, respectively. In *A. arguta* AA, GPs ranged from 8% to 67% after 3–5 weeks of CS. Maghdouri et al. [17] further reported that *A. deliciosa* DA hexaploid genotypes required longer stratification durations for maximum GPs, whereas diploid *A. chinensis* CK1 genotypes required only 4 weeks of CS for optimal germination.

Additionally, Asakuro and Hoshino [33] found that hybrid seeds of *Actinidia* species with varying ploidy levels exhibited different viability rates. They reported that hybrid seeds obtained from crosses involving different ploidy levels had notably low viability and germination rates. Esfandiari et al. [10] also observed that CS improved germination rates in the tested *Actinidia* species, with *A. arguta* achieving 35.6%, *A. chinensis* 1.9%, and *A. deliciosa* 0%. However, CS alone did not significantly enhance germination in most species. Germination performance varied among different kiwifruit genotypes, primarily due to differences in their chilling requirements. Sekhukhune et al. [16] reported that 37 days of CS at 4 °C induced 16% germination in *A. chinensis* CS seeds under in vitro conditions but had no effect on *A. arguta* AA genotypes. Similarly, Maghdouri et al. [17] suggested that variations in germination rates among kiwifruit genotypes could be attributed in their ploidy levels.

##### Gibberellic Acid

The maximum GP of 64% was recorded at a GA_3_ concentration of 2000 ppm, while the minimum GP of 8% was observed at a concentration of 1500 ppm. Çelik et al. [9] reported that seeds treated with 6000 ppm GA_3_ had the minimum GP (35.56%), whereas those treated with 2000 ppm GA_3_ achieved the maximum GP (43.94%). Esfandiari et al. [10] observed that different kiwifruit species, including *A. arguta* Planch. Ex Miq. and *A. valvata*, did not require CS, but instead demonstrated improved germination after being soaked in GA_3_ for 24 h, with maximum GPs of 38.7% and 29.0%, respectively. These species are classified as morphologically dormant, meaning their seeds are dispersed with undeveloped or partially grown embryos that need to grow before germination can occur [34]. Extending the after-ripening period allows the embryo to develop inside the seed before germination begins. Plant growth regulators, particularly abscisic acid (ABA) and gibberellins (GAs), play key roles in regulating seed dormancy and germination [35]. Abscisic acid inhibits germination, while GA suppresses ABA and promotes germination [35]. Although Esfandiari et al. [10] did not measure embryo growth in *A. arguta* and *A. valvata*, their findings suggest morphological dormancy, as these species did not reach the highest germination rates in fresh control tests but responded positively to water or GA_3_ soaking for 24 h. Seeds with morphological dormancy may also exhibit physical dormancy [11]. Generally, if embryo growth and radicle emergence occur within approximately 30 days under suitable conditions, the seeds are considered to have only morphological dormancy [11]. Additionally, Esfandiari et al. [10] reported a maximum GP of 15% in *A. chinensis* seeds treated with GA_3_.

##### Combination of Cold Stratification and GA_3_

In this study, however, combinations of CS and GA_3_ treatment enhanced germination performance. The maximum variable GP of 80% was recorded at the lowest GA_3_ concentration of 500 ppm, while the minimum level of variable GP of 36% was observed at 2500 ppm GA_3_ after 28 days of cold CS. Similarly, extending CS to 42 days followed by GA_3_ treatment also improved germination, with a minimum GP of 32% at 2000 ppm GA_3_ and a maximum variable GP of 72% at 500 ppm GA_3_. These findings indicate that seed germination parameters were significantly influenced by CS duration. Esfandiari et al. [10] also reported optimal germination results when CS was combined with GA_3_. Germination performance varied depending on the kiwifruit species and the duration of CS. Sekhukhune et al. [16] found that in *A. arguta*, 42 days of stratification at 4 °C followed by GA_3_ treatment resulted in an optimal GP of 99%, with a minimum germination time (MGT) of 27 days at GA_3_ concentrations of 1563 and 2120 ppm, respectively. In *A. chinensis* seeds, similar treatments yielded an optimal GP of 45% and an MGT of 23 days at GA_3_ concentrations of 998 and 1629 ppm, respectively.

If seed germination takes longer than 30 days and requires additional dormancy-breaking treatments such as stratification and GA_3_, the seeds are considered to exhibit morphophysiological dormancy [11]. In this study, *A. chinensis* demonstrated a classic case of morphophysiological dormancy, as its seeds required a combination of CS and GA_3_ to overcome dormancy and successfully germinate, resulting in more efficient and faster production of kiwifruit seedlings. In addition to determining the treatments that resulted in the maximum GP, further evaluation was conducted to assess the germination rate by calculating the MGT. The most effective dormancy-breaking treatment was characterized by the highest GP and the lowest MGT. However, the control exhibited the shortest MGT, suggesting that some of the seeds were non-dormant. Treatments that led to an increase in GP while simultaneously reducing MGT likely had a positive effect on breaking seed dormancy. Many researchers suggest that dormancy is a key factor in prolonging germination time [11]. Previous studies have reported that stratification treatments enhance germination rates and reduce germination time [14]. Extended germination periods often indicate seed dormancy, which can be alleviated through various pre-sowing treatments. Generally, when dormancy is broken, germination rates improve due to the elimination of dormancy-related inhibitors, and a strong correlation can exist between the applied germination treatment and the resulting germination rate [16]. In conclusion, our findings indicate that CS alone had a minimal impact on germination. However, CS for 28 days followed by GA_3_ treatment significantly (*p* ≤ 0.05) improved GP in *A. chinensis*. Additionally, treatments with GA_3_ alone and CS for 42 days followed by GA_3_ also proved to be effective in enhancing the germination of *A. chinensis*.

### 3.2. Seedling Development, Shoot Induction and Multiplication

After seed germination, effective plant growth and development are crucial for producing healthy and vigorous seedlings in a short period. Strong and healthy seedlings contribute to early economic fruiting in vineyards and ensure sustainable production. Therefore, this study also focused on investigating seedling development and multiple shoot induction in *A. chinensis* using the plant growth regulator BAP in vitro.

#### Effect of Explant Type and MS Media Supplemented with BAP on Shoot Multiplication

In this study, multiple shoot induction in *A. chinensis* was performed through in vitro shoot culture using apical and basal shoot explants from in vitro-grown seedlings. These explants were cultured on MS medium supplemented with 2.2 µM and 4.4 µM BAP to stimulate shoot multiplication. After four weeks on MS medium with 2.2 µM BAP, no axillary shoot proliferation was observed in apical shoot explants. In contrast, basal shoot explants produced 2–3 axillary shoots, along with tendrils and callus formation at the base, under the same conditions. The basal shoot explants were more responsive than apical shoot explants. Explant type is a factor that can affect shoot proliferation in a number of species, where basal explants have been found to produce more shoots than apical explants [36]. The increased shoot regeneration in basal shoots may be attributed to the absence of apical dominance, as compared to apical shoot explants.

The highest number (3–4) of multiple shoots was attained from these basal shoot explants after subculture (10–12 weeks) to the same culture medium. In contrast, only elongation and rooting occurred in apical shoot explants, without axillary shoot induction, after subculture. A similar response was observed for both types of explants of *A. chinensis* when cultured on MS medium with 4.4 µM BAP. However, the higher concentration of BAP affected the morphological characteristics of the regenerated plantlets, which had shorter stems, and smaller, narrower leaves compared to those derived from 2.2 µM BAP. Similarly, Akbaş et al. [13] found the best results with lower BAP concentrations in germinated stem explants with apical buds of *A. deliciosa*. Their study showed that shoot production had a tendency to decrease as BAP concentrations increased up to 4.0 mg/L, with the best results obtained at 0.5 mg/L BAP, producing 4.7 ± 1.08 shoots per explant after 4 weeks of culture. A similar pattern was observed in the color of the shoots; those grown at 0.5 and 1.0 mg/L BAP produced healthy, green shoots, while those grown at higher concentrations (2.0 and 4.0 mg/L BAP) exhibited a yellowish-green color. Marino and Bertazza [37] found that BAP caused hyperhydricity in older leaves. Similarly, Martini et al. [38] observed that at the multiplication stage, higher BAP concentrations reduced explant response, resulting in fewer shoots with shorter lengths and a higher occurrence of hyperhydration. They found that the highest multiplication index, associated with normal shoots, was achieved with either PGR-free medium or low BAP concentrations. According to Trigiano [39], although higher cytokinin concentrations promote shoot proliferation, the shoots produced are often smaller and may exhibit hyperhydricity symptoms.

### 3.3. Root Induction, Acclimatisation and Transplanting of Plantlets

Rooting of elongated shoots on MS without PGRs was efficient in our study. Regenerated plantlets derived from both apical and basal shoot explants were successfully acclimatised under a controlled environment at 24 ± 2 °C and 16 h photoperiod of 150–200 µmol m^−2^ s^−1^ light intensity. Akbaş et al. [13] observed that the shoots developed under in vitro conditions were rooted on MS medium with 1.0 mg L^−1^ α-napthaleneaceticacid after 3 weeks of culture. However, in their study, other treatments resulted in poor root development (0.5 and 2.0 mg L^−1^). Well-rooted plantlets were successfully adapted to in vivo conditions. According to Trigiano [39], usually, the rooting stage of herbaceous plants can be achieved on medium in the absence of auxins. However, with many woody species, the addition of an auxin (IBA or NAA) in Stage III medium is required to enhance adventitious rooting. However, rooting was successful on MS without PGRS.

## 4. Materials and Methods

### 4.1. Plant Material

*Actinidia chinensis* seeds were extracted from physiologically matured fruits harvested from uniform, healthy and mature kwifruit plants under commercial cultivation at Nooyenskopje Farm at Magoebaskoof (23°53′13″ S, 29°56′13″ E), Tzaneen, in Limpopo Province of South Africa.

### 4.2. Sterilisation of Seeds

Extracted seeds of *A. chinensis* were first washed with soapy water, rinsed under running tap water for 5–10 min (min) and then followed by a rinsing with distilled water (dH_2_O). Seeds were surface sterilised in 70% ethanol for 1 min and then in 50% commercial bleach JIK^®^ (Waterfall City, Midrand, South Africa) solution containing 1.75% active sodium hypochlorite (NaOCl) with a few drops of Tween 20 for 30 min. After each sterilisation procedure, seeds were rinsed few times with sterile dH_2_O to remove the disinfectant solution.

### 4.3. Culture Media

Full-strength Murashige and Skoog [40] medium (MS) containing 3% sucrose (*w*/*v*) and 0.3% (*w*/*v*) gelrite was used for in vitro germination experiments. The seeds were also cultured on sterile moist filter paper (Whatman No. 1) bridges in Sigma^®^ (St. Louis, MO, USA) culture bottles. All aseptic procedures (media preparation, seed sterilisation and inoculation) were conducted in a plant tissue culture laboratory under a plant tissue culture laminar flow cabinet.

### 4.4. Treatment of Seeds and Germination

#### 4.4.1. Cold Stratification

Sterilised seeds of *A. chinensis* were aseptically cultured on sterile moist filter paper bridges and cold stratified (CS) at 4 °C for 28 and 42 days. The durations of CS were selected based on the preliminary results from in vivo germination of *A. chinensis*. Non-stratified (NS) seeds were used as controls. Batches of CS and NS seeds from *A. chinensis* were then germinated on plant growth regulator-free (PGR-free) MS medium and on sterile filter paper bridges moistened with dH_2_O. All seed cultures were kept under controlled growth conditions for germination.

#### 4.4.2. Gibberellic Acid (GA_3_)

An initial study was conducted, where NS seeds of *A. chinensis* were imbibed in culture bottles for 24 h in different GA_3_ concentrations (100, 1000 and 2000 ppm) on an orbital shaker (50 rpm). Seeds imbibed in dH_2_O and dry seeds (no imbibed) served as controls. After imbibition, seeds from all GA_3_ treatments and controls were surface sterilised as described above and were aseptically inoculated on PGR-free MS media for germination. In another experiment, NS sterilised seeds were aseptically inoculated on sterile moist filter paper bridges placed in Sigma^®^ culture bottles and dipped in solutions containing five GA_3_ concentrations (500, 1000, 1500, 2000 and 2500 ppm). The seed cultures were kept under controlled growth conditions. Germinated seeds from these treatments were transferred into PGR-free MS media for seedling development under controlled growth conditions.

#### 4.4.3. Combination of Cold Stratification and GA_3_

Sterilised and CS seeds for 28 and 42 days were aseptically inoculated on filter paper bridges in Sigma^®^ culture bottles containing five GA_3_ solutions (500, 1000, 1500, 2000 and 2500 ppm). The seed cultures were also kept under controlled growth conditions. Germinated seeds from these treatments were transferred into PGR-free MS media for seedling development.

### 4.5. Seedling Development

After 32 days, germinated seeds cultured on the MS medium from the combined stratification durations (28 and 42 days) and 500 ppm GA_3_-treated seeds on the filter paper-bridges were gently removed under sterile conditions and transferred onto the plant growth regulator (PGR)-free MS medium and MS media supplemented with 2.2 µM and 4.4 µM BAP for seedling development. The concentrations of PGR-BAP were selected based on the preliminary results from seedling development of *A. chinensis*.

### 4.6. Seed and Seedling Culture Growth Conditions

All in vitro seed and seedling cultures were kept in a growth room at 24 ± 2 °C and 16 h photoperiod of 50–60 µmol m^−2^ s^−1^ light intensity for germination and seedling development.

### 4.7. Explant Preparation

In vitro-developed seedlings of *A. chinensis* were used as an aseptic explant source for shoot multiplication culture. Shootlet explants were excised from the in vitro raised seedlings, which were dissected into two parts, namely apical shoot explants (+A) and basal shoot explants (−A).

### 4.8. Shoot Induction and Multiplication

Apical (+A) and basal (−A) shoot explants from seedlings were then cultured in MS medium supplemented with 2.2 µM and 4.4 µM BAP for shoot induction and multiplication for 4 weeks. Explants cultured on PGR-free MS medium were used as a control. All cultures were kept in a culture room at 24 ± 2 °C and 16 h photoperiod of 50–60 µmol m^−2^ s^−1^ light intensity.

### 4.9. Root Induction, Acclimatisation and Transplanting of Plantlets

The in vitro multiplied shoots were excised from the mother plant and rooted on a PGR-free MS medium. Successfully rooted plantlets were removed gently from the culture medium and carefully washed with dH_2_O to remove excess agar. Plantlets with developed adventitious roots were then transferred to 5 cm diameter plastic pots filled with moist vermiculite and kept under controlled conditions at 24 ± 2 °C and 16 h photoperiod of 150–200 µmol m^−2^ s^−1^ light intensity. The plantlets were covered with transparent plastic bags, which were gradually opened daily until complete removal to ensure adaptation of plants from in vitro to in vivo conditions. Once the plantlets had fully acclimatised, they were transferred to a greenhouse for further hardening and plant development in vivo.

### 4.10. Experimental Design

All seed germination treatments (CS and GA_3_) and the controls were arranged in a completely randomized design (CRD) with 5 replications, each consisting of 5 seeds (n = 25). Similarly, all seedling treatments (2.2 µM and 4.4 µM BAP) were arranged in a CRD, with each treatment replicated 5 times, with 3 explants per culture vessel.

### 4.11. Data Collection and Analysis

#### 4.11.1. Seed Germination

Radicle emergence from the testa was considered as an indicator for successful germination. In all treatments and the controls, the numbers of germinated seeds were recorded every 7 days until no further germination was observed. Mean germination time (MGT) was calculated using the following equation of Ellis and Roberts [41]:MGT = ΣΣ Dn/ΣΣ n 
where n is the number of seeds that germinate on day D, and D is the number of days from initiation to completion of the germination process. The germination percentage (GP) was calculated as described in the Association of Official Seeds Analysts [42] using the following equation:GP = Germinated seeds/Total number of seeds × 100

Fisher’s least significant difference test was used to separate the means at the probability level of 5%. Germination variables with significant (*p* ≤ 0.05) treatment means were subjected to lines of the best fit.

#### 4.11.2. Shoot Culture

After 21 days of incubation under controlled conditions, the numbers of newly sprouted axillary shoots were recorded weekly. The number of surviving plantlets after the acclimatisation in vivo was recorded and the survival percentage was calculated.

## 5. Conclusions

The seed dormancy of *A. chinensis* was overcome by a combination of CS at 4 °C for 28 days and pretreatment of the seed with 500 ppm GA_3_. A protocol for in vitro regeneration of *A. chinensis* was established that included the following steps: seed germination on moist filter paper bridges, transfer of the germinated seeds to PGR-free MS medium for development; seedling multiplication on 2.2 µM BAP; and rooting on PGR-free MS medium. In vitro regenerated plantlets of *A. chinensis* were successfully acclimatised for 3–4 weeks in vivo inside plastic pots filled with moistened vermiculite by gradual exposure of the plants to lower humidity attaining up 70% survival. This study successfully developed a micropropagation protocol for *A. chinensis*, demonstrating its potential for efficient mass propagation and emphasizing the importance of optimizing BAP concentrations to improve plantlet quality. The findings contribute to the growth of the kiwifruit industry in South Africa by laying the groundwork for commercial cultivation practices.

## Figures and Tables

**Figure 1 plants-14-00939-f001:**
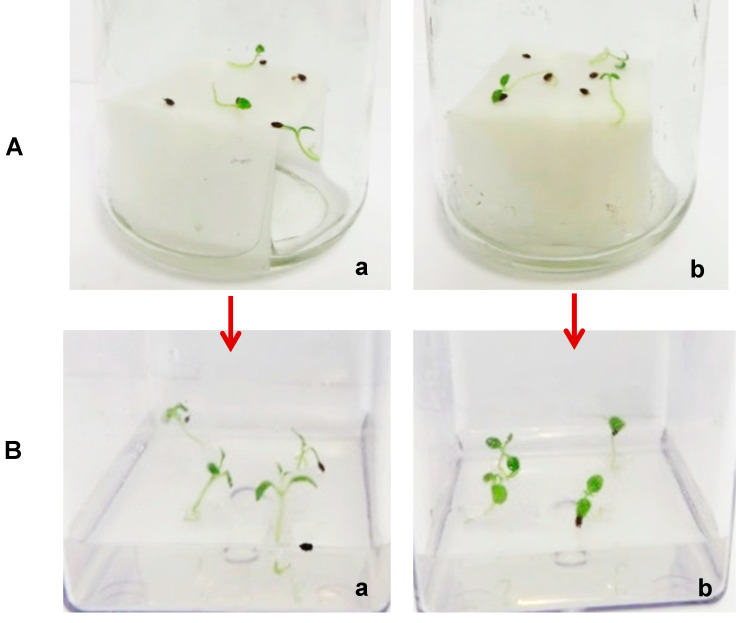
Examples of in vitro culture of *Actinidia chinensis* from different treatments: (**a**) combination of 28 days with 500 ppm GA_3_; (**b**) combination of 42-day CS with 500 ppm GA_3_; (**A**) seeds germinated on filter paper bridges; (**B**) in vitro developed seedlings after subculturing of germinated seeds into full-strength PGR-free MS medium for 4 weeks.

**Figure 2 plants-14-00939-f002:**
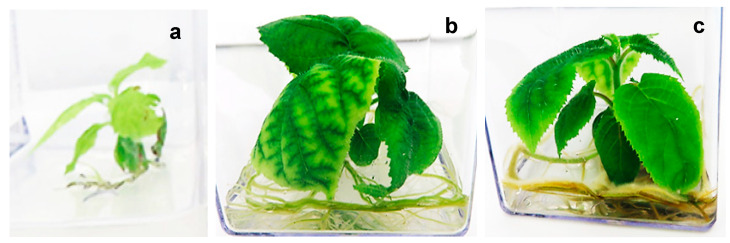
Effect of BAP on in vitro seedling development of *Actinidia chinensis* on MS culture media: (**a**) without BAP; (**b**) supplemented with 2.2 µM BAP; (**c**) supplemented with 4.4 µM BAP.

**Figure 3 plants-14-00939-f003:**
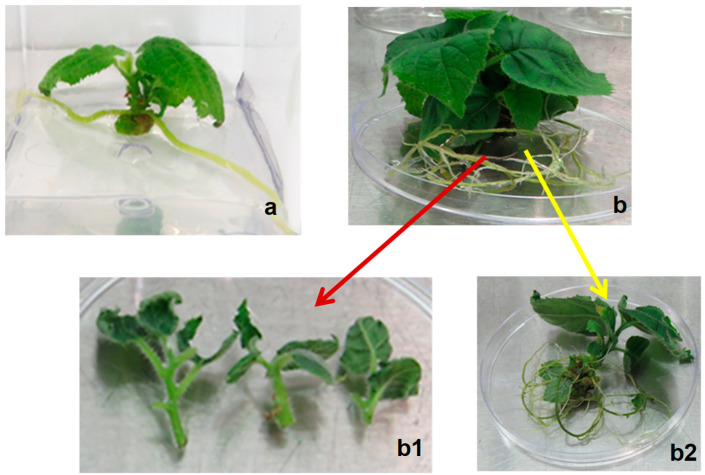
In vitro shoot multiplication of *Actinidia chinensis* on MS media supplemented with 2.2 µM BAP showing regeneration of plantlets after 12 weeks from (**a**) apical shoot explants; (**b**) basal shoot explants; (**b1**) primary axillary shoots obtained from the plantlet; (**b2**) rooted plantlet used for acclimatisation.

**Figure 4 plants-14-00939-f004:**
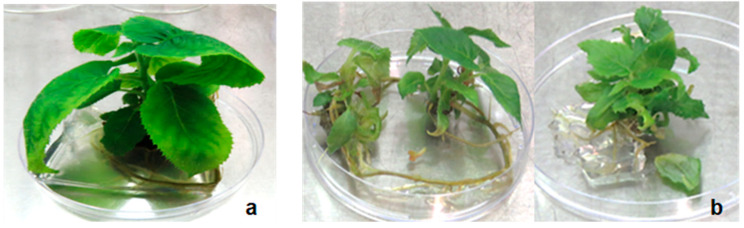
In vitro shoot multiplication of *Actinidia chinensis* on MS media supplemented with 4.4 µM BAP showing regenerated plantlets after 12 weeks from (**a**) apical shoot explants; (**b**) three plantlets from primary axillary shoots and a basal shoot explant.

**Figure 5 plants-14-00939-f005:**
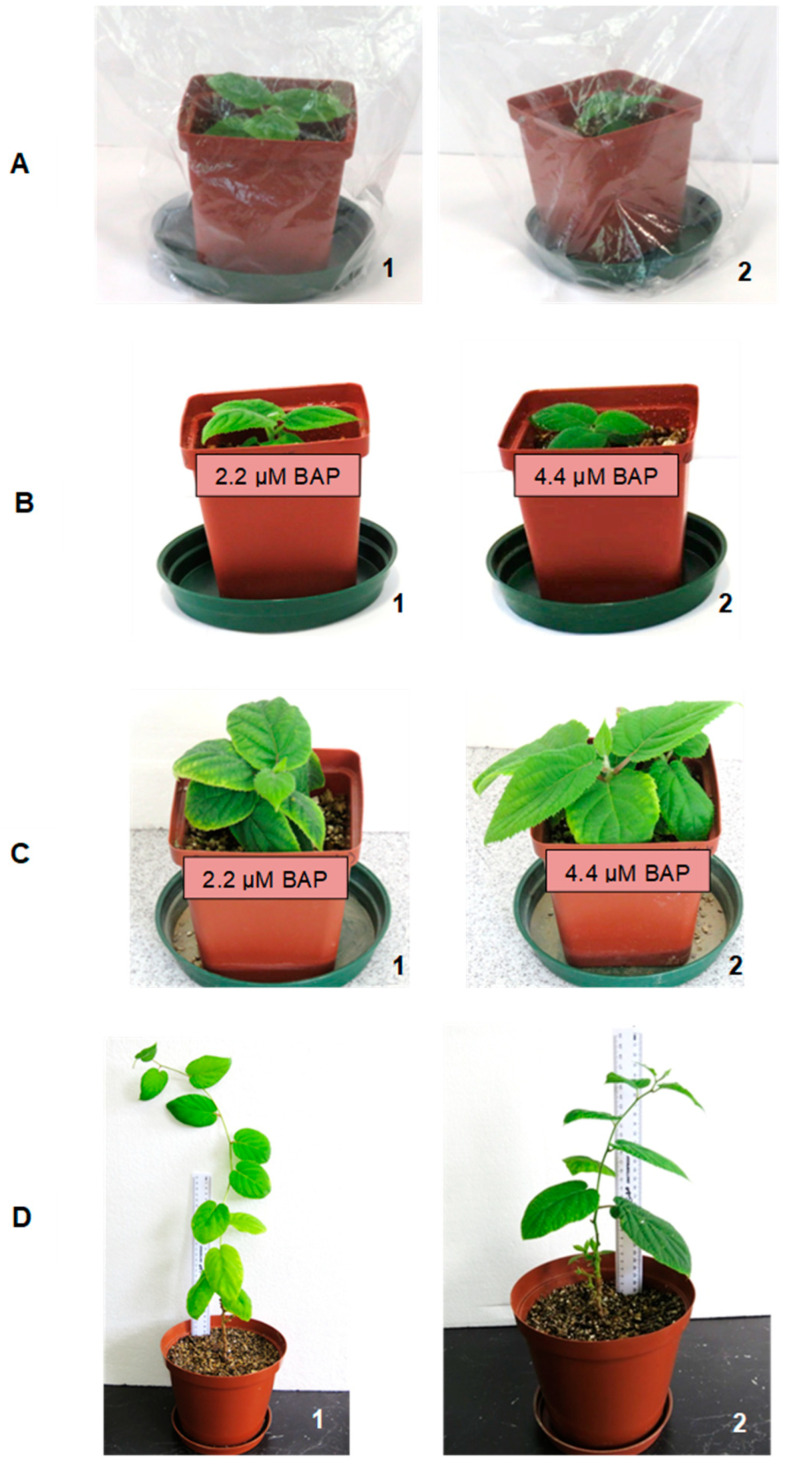
Acclimatisation of in vitro-regenerated plants of *Actinidia chinensis* from different BAP treatments: (1) 2.2 µM BAP; (2) 4.4 µM BAP: (**A**) Plantlets from in vitro culture transplanted to pots filled with vermiculite; (**B**) plants two weeks after the removal of plastic bags; (**C**) acclimatised plants (after 5 months); (**D**) fully acclimatised plants (11 months) ready to be transferred to a glasshouse.

**Table 1 plants-14-00939-t001:** Mean sum of squares for germination percentage and mean germination time (days) of GA_3_ treated *Actinidia chinensis* seeds in vitro.

Source	DF	Germination Percentages		Mean Germination Time (Days)	
		SS	%	SS	%
Treatment	5	16,280	52 *	2491.6	49 *
Error	24	15,200	48	2590.8	51
Total	29	31,480	100	5082.5	100

* Significant at *p* ≤ 0.05.

**Table 2 plants-14-00939-t002:** Effect of GA_3_ on germination percentage and mean germination time of *Actinidia chinensis* seeds in vitro.

Treatment	Germination Percentage	Mean Germination Time
0	4.00 ^b^	4.20 ^c^
500	56.00 ^a^	30.22 ^a^
1000	48.00 ^a^	19.88 ^ab^
1500	8.00 ^b^	14.00 ^bc^
2000	64.00 ^a^	27.52 ^ab^
2500	48.00 ^a^	27.30 ^ab^

Means in columns with the same letter were not significantly (*p* ≤ 0.05) different according to Fisher’s least significant difference test. The letters ^a^, ^b^ and ^c^ indicate significance differences between means.

**Table 3 plants-14-00939-t003:** Mean sum of squares for germination percentage and mean germination time (days) of 28-day cold-stratified and GA_3_-treated *Actinidia chinensis* seeds in vitro.

Source	DF	Germination Percentages		Mean Germination Time	
		SS	%	SS	%
Treatment	5	11,680.0	43 *	1115.27	42 *
Error	24	15,200.0	57	1515.47	58
Total	29	26,880.0	100	2630.73	100

* Significant at *p* ≤ 0.05.

**Table 4 plants-14-00939-t004:** Effect of 28-day cold stratifications followed by GA_3_ on germination percentage and mean germination time of *Actinidia chinensis* seeds in vitro.

Treatment	Germination Percentage	Mean Germination Time
0	20.00 ^c^	7.00 ^b^
500	80.00 ^a^	18.94 ^a^
1000	56.00 ^ab^	18.86 ^a^
1500	68.00 ^ab^	22.20 ^a^
2000	52.00 ^abc^	26.14 ^a^
2500	36.00 ^bc^	23.34 ^a^

Means in columns with the same letter were not significantly (*p* ≤ 0.05) different according to Fisher’s least significant difference test. The letters ^a^, ^b^ and ^c^ indicate significance differences between means.

**Table 5 plants-14-00939-t005:** Mean sum of squares for germination percentage and mean germination time of 42-day cold-stratified and GA_3_-treated *Actinidia chinensis* seeds in vitro.

Source	DF	Germination Percentages		Mean Germination Time	
		SS	%	SS	%
Treatment	5	11,360.0	38 *	1582.59	42 *
Error	24	18,240.0	62	2157.64	58
Total	29	29,600.0	100	3740.23	100

* Significant at *p* ≤ 0.05.

**Table 6 plants-14-00939-t006:** Effect of 42-day cold stratifications followed by GA_3_ on germination percentage and mean germination time of *Actinidia chinensis* seeds in vitro.

Treatment	Germination Percentage	Mean Germination Time
0	8.00 ^c^	2.80 ^b^
500	72.00 ^a^	18.80 ^a^
1000	52.00 ^ab^	16.70 ^a^
1500	36.00 ^bc^	27.30 ^a^
2000	32.00 ^bc^	15.16 ^ab^
2500	40.00 ^abc^	18.76 ^a^

Means in columns with the same letter were not significantly (*p* ≤ 0.05) different according to Fisher’s least significant difference test. The letters ^a^, ^b^ and ^c^ indicate significance differences between means.

## Data Availability

Data of the whole study can be accessed from the authors upon request.

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
