# Peer review of "In Vitro Seed Germination, Seedling Development, Multiple Shoot Induction and Rooting of Actinidia chinensis"

_plants, 2025, doi:10.3390/plants14060939_

Round 1
Reviewer 1 Report
Comments and Suggestions for Authors
The manuscript title “In vitro seedling development, multiple shoot induction and rooting of Actinidia chinensis” is conducted well and has scientific worth. The authors have created a need for effective commercial protocols for the mass propagation of A. chinensis.
Actinidia chinensis fruit is a globally significant crop valued for its nutritional, medicinal, and ornamental properties. In South Africa, the rising local and international demand for kiwifruit has prompted the introduction of new plant species in sub-mountainous areas, where the soil and climate are more conducive to intensive kiwifruit production than in lowland regions.
Summary of this study:
This study presents an optimized micropropagation protocol for A. chinensis, aimed at enhancing seedling development and multiple shoot induction. In vitro shoot culture was conducted using apical and basal shoot explants from in vitro-produced seedlings, cultured on Murashige and Skoog (MS) media supplemented with 2.2 µM and 4.4 µM 6-Benzylaminopurine (BAP) for shoot multiplication.
- Key Findings:
- Axillary shoot proliferation was not observed in apical shoot explants after 4 weeks on MS medium with 2.2 µM BAP, while basal shoot explants produced 2–3 axillary shoots, tendrils, and callus at the base.
- The highest number of multiple shoots (3–4) was achieved from basal shoot explants after subculturing (10–12 weeks) on the same medium.
- Apical shoot explants only exhibited elongation and rooting without axillary shoot induction upon subculture.
- Regenerated plantlets from both explant types were successfully acclimatized in a controlled environment at 24±2°C with a 16-hour photoperiod and light intensity of 150–200 µmol m-2s-1.
- A similar response was noted for both explant types on MS with 4.4 µM BAP; however, the higher concentration resulted in regenerated plantlets with shorter stems and smaller, narrower leaves compared to those derived from 2.2 µM BAP.
- Conclusion: The study successfully establishes a micropropagation protocol for A. chinensis, highlighting the potential for effective mass propagation and the importance of optimizing BAP concentrations to enhance plantlet quality. This research contributes to the expansion of the kiwifruit industry in South Africa by providing a foundation for commercial cultivation practices.
This manuscript has some major issues that need to be addressed.
Comments for authors are as follows:
1- Lines 39, 43, 49, and so on….: please do italic all the scientific names.
2- Lines 42, 44, and so on…. References are not cited according to the Plants journal format.
3- Line 138 and 139: write the full names of each abbreviation when they appear first time in the text.
4- Combine the figure 1 and 2.
5- Lines 157-158: “had well established roots, thicker stems with bigger leaves” how longer roots? In mm or cm?? how much thicker stems?? Add this data and make a graph to show the significant differences etc…. how much folds or percentage root length increased in figure 3B as compared to figure 3A??? Just showing the phenotype seems incomplete. Also add other data for other figures such as how many roots or shoots or their lengths and number of leaves etc….
6- Line 165-166: Where is the legends of figure 3C?
7- Add scale bar in each figure.
8- Figure 4 and 5: move each step figure to supplementary material and only show the most important or final stage of figures in the main manuscript.
9- Line 256: “b) 4.4 µM BA:” it is BAP?
10- “The sterilisation and cold stratification (CS) of the seeds was done according to a study by Sekhukhune et al. (2018).” Add description of this method, listing a reference is not enough. And the spelling of sterilization should be revised.
11- Line 380: statistical analysis should be used just counting taking phenotype pictures is not enough to publish a paper.
Author Response
Comment 1- Lines 39, 43, 49, and so on….: please do italic all the scientific names.
Response 1- [Agreed]. Thank you for pointing this out. We agree with this comment. Therefore, I have used italics for all the scientific names in all the document.
Comment 2- Lines 42, 44, and so on…. References are not cited according to the Plants journal format.
Response 2- [Agreed]. Thank you for pointing this out. We agree with this comment. Therefore, we have cited references according to the Plants journal format.
Comment 3- Line 138 and 139: write the full names of each abbreviation when they appear first time in the text.
Response 3- [Agreed]. Thank you for pointing this out. We agree with this comment. Therefore, we have written full names of each abbreviation appearing for the first time.
Comment 4- Combine the figure 1 and 2.
Response 4- [Agreed]. Thank you for pointing this out. We agree with this comment. Therefore, figure 1 and 2 is combined.
Comment 5- Lines 157-158: “had well established roots, thicker stems with bigger leaves” how longer roots? In mm or cm?? how much thicker stems?? Add this data and make a graph to show the significant differences etc…. how much folds or percentage root length increased in figure 3B as compared to figure 3A??? Just showing the phenotype seems incomplete. Also add other data for other figures such as how many roots or shoots or their lengths and number of leaves etc…
Response 5-[Agreed]. Thank you for pointing this out. We agree with this comment. However, data was only collected for germination parameters.
Comment 6- Line 165-166: Where is the legends of figure 3C?
Response 6- [Agreed]. Thank you for pointing this out. We agree with this comment. Therefore, legend for figure 3C is provided.
Comment 7-Add scale bar in each figure.
Response 7-[Agreed]. Thank you for pointing this out. We agree with this comment. Therefore,Scale bar is added.
Comment 8- Figure 4 and 5: move each step figure to supplementary material and only show the most important or final stage of figures in the main manuscript.
Response 8- [Agreed]. Thank you for pointing this out. We agree with this comment. Therefore, we have showed the most important or final stage of Figure 4 and 5 in the main manuscript.
Comment 9- Line 256: “b) 4.4 µM BA:” it is BAP?
Response 9- [Agreed]. Thank you for pointing this out. We agree with this comment. Therefore, BA is replaced with BAP.
Comment 10- “The sterilisation and cold stratification (CS) of the seeds was done according to a study by Sekhukhune et al. (2018).” Add description of this method, listing a reference is not enough. And the spelling of sterilization should be revised.
Response 10: [Agreed]. Thank you for pointing this out. We agree with this comment. Therefore, description of the treatment of germination was added.
Comment 11: Line 380: statistical analysis should be used just counting taking phenotype pictures is not enough to publish a paper.
Response 11: [Agreed]. Thank you for pointing this out. We agree with this comment. Therefore, statistical analysis is used.
Reviewer 2 Report
Comments and Suggestions for Authors
Comments and Suggestions for Authors
This study aimed to create an optimized micropropagation protocol for Actinidia chinensis, an important crop with greater economic significance due to its nutritional, medicinal and orna-mental values. The protocol is based on Actinidia chinensis seeds germinated directly on MS medium without PGR and sequential seedling development, induction of multiple shoots and rooting under the influence of BAP.
The manuscript is constructed according to requirements of “Plants”. The applied research methods are suitable for achieving the research objective. The results are well illustrated. However, there is no computational processing of data from various tests, such as seed germination percentage, closed and produced seedlings, rooted and adapted seedlings, etc.
Additional remarks and suggestions can be made:
Introduction
- The first part of this section /lines 39-60/ needs redaction: 1. The first sentence /lines 39-42/ is too long and confused. Мy suggestion for it is: ‘Kiwifruit (Actinidia chinensis Planch) commonly referred as “Chinese kiwifruit’’, is a woody perennial, dioecious, medicinal plant belonging to Actinidia Lindl. genus (Actinidiaceae) comprising 75 species and about 125 taxa native to China (Li et al., 2006)’ 2. The sentence in lines 46-49: “The fruit has … (Ferguson, 1999)” have to be omitted. 3. The last sentence /lines 57-60/ must finish by “…with more than 4,348,011 metric tonnes of global kiwifruit production” /line 59/.
- The next part of this section /lines 62-92/ must be shortened to:”The variety of A. delicosa ‘Hayward’ is primarily cultivated and is targeted at local and export markets to the Southern African Development Community (SADC) countries (Hoyi, 2014). Thus, the call to expand kiwifruit production in South Africa and globally calls for the development of commercial protocols for mass propagation of A. chinensis that can be used to expand kiwifruit production.” and should be placed on line 59 as a continuation of the part above.
- The sentence in lines 103-104 should be redacted as: “In vitro culture is another method of mass plant production that can be used for plants that are difficult to propagate conventionally.”, and the entire rest part of this paragraph /up to line 115/ should be removed because it contains well-known facts about what in vitro culture is.
- - "In vitro" and "A.chinensis" should be in italics
Results
- The received data in different steps of micropropagation /germination, shoot proliferation, seedlings routing and adaptation/ carried out should be presented beside in figures /photographs/, in tables and/or graphs on which base can be made comparative analysis to define the more effective culture conditions.
Materials and Methods
- Specify why BAP is chosen as PGR – on the base of preliminary studies with other Actinidia species or other cause?
Discussion
- As is mentioned in “Introduction”, the seed dormancy is the cause of poor investigations of in vitro multiplication of A.chinensis by seed germination. The seed dormancy overcoming is one of the achievements of present study. Thus, the dormancy in studied species and manner of its overcoming should be discussed.
- The part in lines 292-324 contain general information about cytokinins and their role in tissue culture, especially since only BAP is used in the study.
Conclusions
- This section should be rewritten. At the present it is a results comment. It should point the advantages of the study and contain an assumption of the protocol established, as: “The seed dormancy of Actinidia chinensis was overcome by cold storage at … 0C and pretreatment of the seed with ..% GA3. A protocol for in vitro regeneration of A. chinensis was established that includes the following steps: seed germination on free of PGRs MS medium; seedlings multiplication on …; rooting on …; plant adaptation ….” The significance and future perspective of the results should also be indicated.
In conclusion, this manuscript is recommended for publication in "Plants" after consideration of the comments made.
Author Response
Introduction
Comment 1
- The first part of this section /lines 39-60/ needs redaction: 1. The first sentence /lines 39-42/ is too long and confused. Мy suggestion for it is: ‘Kiwifruit (Actinidia chinensis Planch) commonly referred as “Chinese kiwifruit’’, is a woody perennial, dioecious, medicinal plant belonging to Actinidia Lindl. genus (Actinidiaceae) comprising 75 species and about 125 taxa native to China (Li et al., 2006)’
Response 1 [Agreed]. Thank you for pointing this out. We agree with this comment. Therefore, redaction was povided as suggested.
Comment 2. The sentence in lines 46-49: “The fruit has … (Ferguson, 1999)” have to be omitted.
Response 2 [Agreed]. Thank you for pointing this out. We agree with this comment. Therefore, the sentence in lines 46-49: “The fruit has … (Ferguson, 1999)” is omitted.
Comment 3. The last sentence /lines 57-60/ must finish by “…with more than 4,348,011 metric tonnes of global kiwifruit production” /line 59/.
Response 3 [Agreed]. Thank you for pointing this out. We agree with this comment. Therefore, the last sentence /lines 57-60/ finished by “…with more than 4,348,011 metric tonnes of global kiwifruit production” /line 59/.
Comment 4- The next part of this section /lines 62-92/ must be shortened to:”The variety of A. delicosa ‘Hayward’ is primarily cultivated and is targeted at local and export markets to the Southern African Development Community (SADC) countries (Hoyi, 2014). Thus, the call to expand kiwifruit production in South Africa and globally calls for the development of commercial protocols for mass propagation of A. chinensis that can be used to expand kiwifruit production.” and should be placed on line 59 as a continuation of the part above.
Response 4- [Agreed]. Thank you for pointing this out. We agree with this comment. Therefore, line 62 is shorted. However, I need clarity please, should it be from line 62-92?
Comment 5 The sentence in lines 103-104 should be redacted as: “In vitro culture is another method of mass plant production that can be used for plants that are difficult to propagate conventionally.”, and the entire rest part of this paragraph /up to line 115/ should be removed because it contains well-known facts about what in vitro culture is.
Response 5 [Agreed]. Thank you for pointing this out. We agree with this comment. Therefore, the sentence in lines 103-104 was redacted as: “In vitro culture is another method of mass plant production that can be used for plants that are difficult to propagate conventionally.”, and the entire rest part of this paragraph /up to line 115/ is removed.
Comment 6 - "In vitro" and "A.chinensis" should be in italics
Response 6 [Agreed]. Thank you for pointing this out. We agree with this comment. Therefore, "in vitro" and "A. chinensis" is italics throughout the manuscript.
Materials and Methods
Comment 1- Specify why BAP is chosen as PGR – on the base of preliminary studies with other Actinidia species or other cause?
Response 1- "The concentrations of PGR-BAP were selected based on the preliminary results from seedling development of A. chinensis." is added.
Discussion
Comment 1- As is mentioned in “Introduction”, the seed dormancy is the cause of poor investigations of in vitro multiplication of A.chinensis by seed germination. The seed dormancy overcoming is one of the achievements of present study. Thus, the dormancy in studied species and manner of its overcoming should be discussed.
Response 1 [Agreed]. Thank you for the comment, we agree with the comment. Therefore, the dormancy in studied species and manner of its overcoming should is discussed.
Comment 2 - The part in lines 292-324 contain general information about cytokinins and their role in tissue culture, especially since only BAP is used in the study.
Response 2 [Agreed]. Thank you for the comment, we agree with the comment. Therefore, the part in lines 292-324 is removed.
Conclusions
Comment 1- This section should be rewritten. At the present it is a results comment. It should point the advantages of the study and contain an assumption of the protocol established, as: “The seed dormancy of Actinidia chinensis was overcome by cold storage at … 0C and pretreatment of the seed with ..% GA3. A protocol for in vitro regeneration of A. chinensis was established that includes the following steps: seed germination on free of PGRs MS medium; seedlings multiplication on …; rooting on …; plant adaptation ….” The significance and future perspective of the results should also be indicated.
Response 1-[Agreed]. Thank you for the comment, we agree with the comment. Therefore, the section is rewritten as suggested and the significance and future perspective of the results is indicated.
Round 2
Reviewer 1 Report
Comments and Suggestions for Authors
Th authors did revisions but still somethings are left. 1- the reference citation formate is wrong. the first reference citation would be [1]. but the authors cited [31] in line 65. please order all the citation according tot he journal.
2- Introduction is still seems to be longer, normally 3-4 paragraph introduction is enough.
3- Lines 730-1042: the discussion section is very long. It looks like a discussion of thesis. Please keep only relavent information in th discussion section. and decrease the length of discussion less than 150 lines. currently, it is 312 lines.
Author Response
Comment: 1- the reference citation formate is wrong. the first reference citation would be [1]. but the authors cited [31] in line 65. please order all the citation according tot he journal.
Response 1- [Agreed] Thank you very much for your comment, we agree with the suggestion, therefore the reference citation was done according to the journal
Comment 2- Introduction is still seems to be longer, normally 3-4 paragraph introduction is enough.
Response 2-[Agreed] Thank you very much for your comment, we agree with the suggestion, therefore the introduction was reduced to 3-4 paragraphs
Comment 3- Lines 730-1042: the discussion section is very long. It looks like a discussion of thesis. Please keep only relavent information in th discussion section. and decrease the length of discussion less than 150 lines. currently, it is 312 lines.
Response 4-[Agreed] Thank you very much for your comment, we agree with the suggestion, therefore the discussion is reduced.
Reviewer 2 Report
Comments and Suggestions for Authors
Comments and Suggestions for Authors
In the revised version of the manuscript, the authors have taken into account some of the recommendations and suggestions made in the previous review. However, there are still parts that need to be improved:
Some remarks can be made regarding the style of expression. For example, the sentences in lines 120-121: Cold stratified seeds for 28 days yielded a maximum of 20% germination percentage (GP). Whereas, cold stratified seeds (42 days) resulted in 8% GP. will be better to combine in one sentence: Seeds cold stratified for 28 days yielded a maximum of 20% germination percentage (GP), while those stratified for 42 days yielded only 8% GP.; In line 135: The lowest variable GPs of 4% will be recorded at the control (Table 2), “will be” have to be changed with “was”
In the tables, the asterisk and letter designations /a, b, c, ab, abc/ must be indicated as a footnote below the tables.
The "Discussion" section remains in its previous version and needs editing, as recommended:
- As is mentioned in “Introduction”, the seed dormancy is the cause of poor investigations of in vitro multiplication of A.chinensis by seed germination. The seed dormancy overcoming is one of the achievements of present study. Thus, the dormancy in studied species and manner of its overcoming should be discussed. even more from the data shown in the tables resulted that the control shows the shortest „Mean germination time“
- The part in lines 292-324 /now lines 312-343/ contains general information about cytokinins and their role in tissue culture not necessary for this study, especially since only BAP is used in the study.
The “Conclusions” also need correction as recommended:
- This section should be rewritten. At the present it is a results comment. It should point the advantages of the study and contain an assumption of the protocol established, as: “The seed dormancy of Actinidia chinensis was overcome by cold storage at … 0C for …. hauars and pretreatment of the seed with ..% GA3. A protocol for in vitro regeneration of A. chinensis was established that includes the following steps: seed germination on free of PGRs MS medium; seedlings multiplication on …; rooting on …; plant adaptation ….” The significance and future perspective of the results should also be indicated.
In conclusion, this manuscript is recommended for publication in "Plants" after consideration of the comments made.
Author Response
Comment 1: Some remarks can be made regarding the style of expression. For example, the sentences in lines 120-121: Cold stratified seeds for 28 days yielded a maximum of 20% germination percentage (GP). Whereas, cold stratified seeds (42 days) resulted in 8% GP. will be better to combine in one sentence: Seeds cold stratified for 28 days yielded a maximum of 20% germination percentage (GP), while those stratified for 42 days yielded only 8% GP.; In line 135: The lowest variable GPs of 4% will be recorded at the control (Table 2), “will be” have to be changed with “was”
Response 1: [Agreed] Thank you very much for the comment, We agree, therefore we have implemented the suggestion.
Comment 2: In the tables, the asterisk and letter designations /a, b, c, ab, abc/ must be indicated as a footnote below the tables.
Response 2:[Agreed] Thank you very much for the comment, We agree, therefore we have added asterisk and letter designations as a footnote below the tables.
Comment 3:
The "Discussion" section remains in its previous version and needs editing, as recommended:
- As is mentioned in “Introduction”, the seed dormancy is the cause of poor investigations of in vitro multiplication of A.chinensis by seed germination. The seed dormancy overcoming is one of the achievements of present study. Thus, the dormancy in studied species and manner of its overcoming should be discussed. even more from the data shown in the tables resulted that the control shows the shortest „Mean germination time“
- The part in lines 292-324 /now lines 312-343/ contains general information about cytokinins and their role in tissue culture not necessary for this study, especially since only BAP is used in the study
Response 3: [Agreed] Thank you very much for the comment, We agree, therefore the dormancy in studied species and manner of its overcoming is discussed. Again line 312-343 is removed and replaced.
Comment 4: This section should be rewritten. At the present it is a results comment. It should point the advantages of the study and contain an assumption of the protocol established, as: “The seed dormancy of Actinidia chinensis was overcome by cold storage at … 0C for …. hauars and pretreatment of the seed with ..% GA3. A protocol for in vitro regeneration of A. chinensis was established that includes the following steps: seed germination on free of PGRs MS medium; seedlings multiplication on …; rooting on …; plant adaptation ….” The significance and future perspective of the results should also be indicated.
Responde 4: [Agreed] Thank you very much for the comment, We agree with the comment, therefore the “Conclusions” is corrected as recommended
Round 3
Reviewer 2 Report
Comments and Suggestions for Authors
Comments and Suggestions for Authors
In this final version of the manuscript, the authors have taken into account all the comments made and it is now suitable for publication in "Plants"
Author Response
Thank you very much for the review